# Gene Functional Networks from Time Expression Profiles: A Constructive Approach Demonstrated in Chili Pepper (*Capsicum annuum* L.)

**DOI:** 10.3390/plants12051148

**Published:** 2023-03-03

**Authors:** Alan Flores-Díaz, Christian Escoto-Sandoval, Felipe Cervantes-Hernández, José J. Ordaz-Ortiz, Corina Hayano-Kanashiro, Humberto Reyes-Valdés, Ana Garcés-Claver, Neftalí Ochoa-Alejo, Octavio Martínez

**Affiliations:** 1Unidad de Genómica Avanzada (Langebio), Centro de Investigación y de Estudios Avanzados del Instituto Politécnico Nacional (Cinvestav), Irapuato 36824, Mexico; 2Departamento de Investigaciones Científicas y Tecnológicas de la Universidad de Sonora, Hermosillo 83000, Mexico; 3Department of Plant Breeding, Universidad Autónoma Agraria Antonio Narro, Saltillo 25315, Mexico; 4Unidad de Hortofruticultura, Centro de Investigación y Tecnología Agroalimentaria de Aragón, Instituto Agroalimentario de Aragón-IA2 (CITA-Universidad de Zaragoza), 50059 Zaragoza, Spain; 5Departamento de Ingeniería Genética, Centro de Investigación y de Estudios Avanzados del Instituto Politécnico Nacional (Cinvestav), Irapuato 36824, Mexico

**Keywords:** gene expression, RNA-Seq, time expression profile, fruit development, *Capsicum*

## Abstract

Gene co-expression networks are powerful tools to understand functional interactions between genes. However, large co-expression networks are difficult to interpret and do not guarantee that the relations found will be true for different genotypes. Statistically verified time expression profiles give information about significant changes in expressions through time, and genes with highly correlated time expression profiles, which are annotated in the same biological process, are likely to be functionally connected. A method to obtain robust networks of functionally related genes will be useful to understand the complexity of the transcriptome, leading to biologically relevant insights. We present an algorithm to construct gene functional networks for genes annotated in a given biological process or other aspects of interest. We assume that there are genome-wide time expression profiles for a set of representative genotypes of the species of interest. The method is based on the correlation of time expression profiles, bound by a set of thresholds that assure both, a given false discovery rate, and the discard of correlation outliers. The novelty of the method consists in that a gene expression relation must be repeatedly found in a given set of independent genotypes to be considered valid. This automatically discards relations particular to specific genotypes, assuring a network robustness, which can be set a priori. Additionally, we present an algorithm to find transcription factors candidates for regulating hub genes within a network. The algorithms are demonstrated with data from a large experiment studying gene expression during the development of the fruit in a diverse set of chili pepper genotypes. The algorithm is implemented and demonstrated in a new version of the publicly available R package “*Salsa*” (version 1.0).

## 1. Introduction

Gene co-expression networks (GCN) [1] are graphs that connect genes (vertices) by lines (edges), which represent the fact that genes were co-expressed in space (cell, tissue, organ, etc.), time (for example, during development), or under particular environmental conditions. In general, graph theory has been found to be useful to analyze biological networks [2], and, in particular, GCN have been used to quantify relationships between co-expression, co-regulation, and gene function [3], and some functional modules have been found to be conserved in very distant species, as in yeast and humans [4].

GCN can be estimated using different statistical approaches, such as, for example, Pearson’s correlation, graphical Gaussian models, Bayesian approaches, or methods based on mutual information [5]. Aside the statistical significance of the relations estimated, the main point of GCN is to summarize information about gene modules associated with particular biological processes, providing novel insights into the system-level understanding of cellular processes [6]. This process could lead to unraveling gene function in crops [7], which, in turn, will allow the efficient application of molecular-aided methods of plant breeding [8].

Nonetheless, when the number of genes and relations in a GCN are large, its plot resembles a “hairy ball” [9], and thus could be almost useless for direct interpretation, given the human limits for the perception of relations [10]. In Appendix A, we present an example of a large GCN.

Beyond a GCN—which will simply show significant co-expression of genes in a single genotype—we must look for gene functional networks (GFN), which will display robust functional relations between genes. An ideal GFN must show causal relationships between genes, which follow an alike time expression profile and are robust in the sense of being independently repeated in various genotypes.

Causality, our first requirement for a GFN, demands a deep knowledge of the process involved, which is rarely found out of model organisms, as *Arabidopsis* for plants or mice for animals. However, gene ontology (GO) [11] groups genes into biological process (BP), and gene orthology permits us to infer genes with the same BP in non-model species, allowing the prediction of GO BP from temporal gene expression patterns, even for non-annotated genes [12]. It appears reasonable to assign causality to well annotated genes that share identical BP in the GO, and we will assume that as a fact. However, we must take into account that the gene to BP relation is not one to one, i.e., a gene is usually annotated in different BPs, forming directed acyclic graphs, and this, as well as the quality of annotations, must be taken into account to avoid mistakes [13].

Asking for time co-expression of genes, our second requirement for a GFN, limits our definition to cases where gene expression was estimated at a grid of increasing times, presenting significant changes in at least one pair of neighbor times. This requirement excludes genes that, even when causally participating in a BP, do not show significant expression changes during the whole interval observed, and, thus, these could be labeled as “housekeeping” genes [14] for the BP of interest. On the other hand, it is axiomatic that genes with highly similar expression profiles are likely to be regulated via the same mechanisms [3,15], and thus this fact is integrated in our GFN definition.

Accuracy, scalability, robustness and reproducibility are important qualities of GFN [16]. GFN can vary depending on the genotype, for example, GFN including WRKY transcription factors in rice show variation between the Indica and Japonica genotypes [17]. In our GFN definition, robustness is evaluated by the number of times that a relation between two genes is independently detected in different genotypes at a fixed false discovery rate (FDR) [18] selected by the researcher. If a gene relation is found only in a single accession or genotype, but absent in all the others, that relation cannot be part of a general GFN for the species of interest, and it is more likely to occur by chance or by particularities of that genotype. In contrast, gene relations that are found in all independent genotypes studied are likely to be causal and must form part of the GFN for the species of interest. We designed here a measure of robustness for gene relations, which allows the researcher to set a level of confidence for the relations to be included in the network (see Section 2).

The main goal of our work was to obtain a method to estimate robust and small GFN modules for specific biological processes, using independent time expression profiles from different genotypes. Such modules can then be linked together into larger networks by the genes shared between them. The novelty of this approach consists in the design and implementation of an algorithm that uses standardized expression profiles (SEPs) [19,20] from independent experiments. This allows us to judge and tune the robustness of the GFN modules, assuring that they show only relations, which are confirmed in a set of representative and independent genotypes. Later, by orderly linking together GFN modules, we avoid the “hairy ball” effect, obtaining a network that is easier to interpret than the ones obtained by methods that estimate a whole network from all genes at once, as is the case with ARACNE [21] and other approaches [22].

A secondary goal of our approach was to formalize an algorithm to estimate transcription factor (TF) candidates to be regulating genes present in a GFN. Briefly, if no TFs are included in a GFN, the algorithm looks for TF genes, which have highly correlated time expression profiles with the genes in the GFN, demanding also that such correlation will be present in various independent genotypes. If found, those TFs are presented as candidates to regulate the gene or genes of interest within the GFN. That method has successfully applied to recover TFs that regulate the *AT3* gene, involved in capsaicin’s biosynthesis in chili pepper [23].

The methods are demonstrated in a collection of transcriptomes obtained from 12 accessions during the development of the chili pepper fruits [19], and they are implemented in a new version of the publicly available R package “*Salsa*” [24].

## 2. Methods

### 2.1. Prerequisites

To apply our method to estimate GFN, the researcher must have genome-wide gene expression profiles, resulting from a time–course experiment in different genotypes of the species of interest. Even when such gene expression profiles could be the result of microarray experiments—as exemplified in [25]—here we assume that the data were obtained with the RNA-Seq technology [26,27], which in general appears to have a better performance for GCN estimation than microarrays [22].

Given that both the genotype [28], as well as the environment [29] and their interaction [30], can influence gene expression, the generality of GFN can be assured only for the diverse genotypes and environments where it was tested. For this reason, our method demands that the expression profiles must be obtained from a set of different genotypes in independent experiments performed under the same environmental conditions. If those requirements are fulfilled, then our method will give robust GFNs with the stringency that can be fixed by the researcher.

### 2.2. Congruent Gene Correlations

Given that our method is based on congruent gene expression profiles, which are repeated in independent genotypes, Figure 1 presents an illustrative example of that case.

Figure 1 presents standardized expression profiles (SEPs) [19,20] for two genes (Kinesin-4 and HMGB prot. 6) in two different chili pepper accessions, “Ancho San Luis”, a domesticated accession, and the wild accession, “Piquín Quéretaro”, with keys “AS” and “QU” in the figure legend. We can see how these two genes present a highly congruent expression behavior within each one of the two independent genotypes (AS and QU), a fact that is reflected in very high values of Pearson’s correlations, 0.998 for AS and 0.960 for QU. Additionally, these correlation coefficients are highly significant, having *p*-values $<6\times 10^{-4}$. In contrast, the two highly correlated gene pairs show distinct and uncorrelated expression profiles in the two different accessions (see Appendix A). We will see that, if correlation between the same pair of genes is found in various independent accessions, this strongly suggests that such a co-expressing gene pair could be indeed functionally related—even when the genes could have different and uncorrelated time expression profiles, as shown in Figure 1.

### 2.3. Robustness of a Relation between Genes

Any method to estimate GFN must give an associated measure of robustness to judge how strong the gene relations displayed in the network are. A now customary method to protect against false positives when many statistical tests are performed is the transformation of *p*-values into *q*-values, which allows the setting of a False Discovery Rate (FDR) threshold [18].

Our method to predict the minimum robustness of a GFN couples the setting of a FDR threshold with the fact that the relation must be found in various independent genotypes. Assume, for example, that we have a total of a=12 independent genotypes, representing a fair sample of the genetic diversity existent in the species of interest, and that we set a maximum FDR of 5%. Then, the probability of finding a fortuitous relation between two genes in a particular genotype will be 5%. However, given that the genotypes are evaluated in independent experiments, we can ask that the relation must be corroborated with the same FDR in various genotypes to be taken into account in the GFN; we can ask, for example, that the relation must be repeatedly found in x=2,3,⋯,a. In Appendix A, we show that the probability of finding the same gene relation independently repeated in *x* genotypes depends on the binomial distribution, and it rapidly decreases when *x* increases, being minimal at x=a. For example, by fixing a FDR of 5%, if we ask that a relation must be found in a minimum of x=5 genotypes, then the probability of that relation being by pure chance is small, approximately 0.0002 or one in 5436. If we want to be more strict, we could ask that the relation must be repeatedly found in the whole set of genotypes, i.e., in the x=12 accessions. In that last case, the probability of that relation being random is very small, $\approx 2.4\times 10^{-16}$; this is less than one in many billions and, for all practical purposes, we could be “almost sure" that the relation is real and not given by chance.

In summary, when a reasonable number of independent genotypes are available, the researcher could obtain robust GFN values, which very likely represent true functional relations between genes.

### 2.4. The “Gene2Gene” Algorithm

The objective of the *Gene2Gene* algorithm is to obtain a GFN. As mentioned before, the algorithm assumes the existence of time expression profiles in a group of genotypes, and it needs as input:The genes of interest, say a set of “*g*” gene identifiers.The genotypes (or accessions) where the estimation will be performed, i.e., a set of “*a*” genotypes.A threshold for the FDR, “*f*”.A threshold for the minimum value of Pearson’s determination coefficient, “$mr^{2}$”.A threshold to eliminate putative regression outliers, “*q*”.The minimum number of genotypes where the gene relation must be found to be reported in the output, “*x*” (x≤a).

Given the input, in a first step, the algorithm estimates and tests all g(g−1)/2 pairs of correlations between the *g* genes of interest in each one of the *a* genotypes. In a second step, the algorithm selects a set of genes that fulfill all criteria given in the input, and then it outputs all the relevant statistics to construct a GFN. The output could be null—when there are no genes that fulfill the input thresholds or a set of structures that define a GFN.

Even when the *Gene2Gene* algorithm has been implemented in R [31] within the package *Salsa* [24], specifically for data from a *Capsicum* experiment [19], it can be adapted to process any set of gene expression profiles. Details of the *Gene2Gene* algorithm are given in Appendix A.

### 2.5. Constructing GFN: A Bottom-Up Approach

One important property of biological networks is their modular structure. In many relevant cases, as in *Arabidopsis* [32], humans [33], or yeast [33], genes within functional modules are densely connected, while, in contrast, connections between modules can be sparse or inexistent.

It is certainly possible to construct a general gene co-expression network with data for all genes in the genome, such as, for example, the one constructed for *Arabidopsis* [32], and then one can detect and order the modularity, as reviewed in [34]. Modularity can be based on different criteria, such as, for example, biological process, cell component, molecular function or metabolic pathway, and modules from different criteria will in general present highly complex and difficult to interpret relations. Additionally, co-expression networks estimated for all genes in the genome will include not identified or not annotated genes, which will complicate interpretation even further.

We suggest that a bottom-up approach, beginning with a careful selection of a relatively small set of genes annotated in a process of interest, and progressing to networks of different modules, i.e., sets of GFN or “Meta Networks” (MN), will be easier to interpret and could lead to relevant and immediate discoveries. Additionally, with our method, the generality (robustness) of each GFN can be statistically evaluated, and the analysis could be performed with different stringency levels.

The same bottom-up approach could be employed with different criteria, but not necessarily with genes that share a known annotation in a given biological process. For example, assume that there is a set of genes that are of special interest for a research group, without being annotated in any particular aspect. In that case, the *Gene2Gene* algorithm can be run with that set of genes to determine if they form a significant gene expression network. See Appendix A.

## 3. Results

### 3.1. Data Analyses

As an example of the application of the *Gene2Gene* algorithm, we present the estimation of GFN for three biological processes (BP). The original data were duplicated RNA-Seq libraries from fruits of 12 genotypes (accessions) of chili pepper (*Capsicum annuum* L.) sampled in each case at seven times of development. These genome-wide data were annotated with GO categories [11], and they were processed to give standardized time expression profiles (SEPs) [19,20]. All data and functions employed here to perform the analyses are implemented in a new version of the publicly available R package “*Salsa*” [24], and details of the computations and results are presented in Appendix A.

#### 3.1.1. GFN for Three Biological Processes (BPs)

We selected the BP “*cell cycle*”, “*reproduction*” and “*response to virus*”, which are abbreviated here as “*celcy*”, “*rep*” and “*vir*”, respectively. The parameters for the *Gene2Gene* algorithm were set to a FDR of 10%, a minimum of ${\hat{r}}^{2}$ of 0.7, and a threshold to eliminate 5% of putative regression outliers. We also asked that the output will give gene relations that were repeated in all the 12 genotypes in the collection. Table 1 summarizes the results obtained for the three BPs.

In Table 1, we can see that the algorithm began with 352 genes that were annotated in the *celcy* BP, but, from all the (352×(351−1))/2 = 61,600 pair correlations tested per genotype, only 81 of them, involving 29 genes, passed the thresholds imposed and were repeated consistently in the full set formed by the 12 genotypes; thus, only approximately 8% of the total set of genes is corroborated to have co-expressions. Even when the algorithm was run with parameters of moderate stringency, i.e., a minimum ${\hat{r}}^{2}=0.7$—which implies a positive value of $\hat{r}\geq 0.83$, and a FDR of 10% ($\hat{q}\leq 0.1$), the actual values of the averages of $\hat{r}$ and $\hat{q}$, presented as $\bar{r}$ and $\bar{q}$ respectively in Table 1, are highly superior to the threshold values set; $\bar{r}=0.9859$ is much larger than the threshold of $\hat{r}\geq 0.83$, while the values of $\bar{q}=0.00729$ is much smaller than the threshold of $\hat{q}\leq 0.1$ asked to filter by FDR. The same behavior on the comparisons between realized values of $\hat{r}$ and $\hat{q}$ can be observed in the other two GFNs, *rep* and *vir* in Table 1, i.e., the average values are far away from the thresholds set. This is a direct result of the fact that we are asking that the correlations must be present in the full set of 12 different genotypes, and this highly stringent criterion is likely to give only highly congruent and truthful relations that happen in domesticated, wild, as well as $F_{1}$ crosses of *Capsicum* genotypes [15,19].

Figure 2, Figure 3 and Figure 4 present the plots for the *celcy*, *rep*, and *vir* GFN, respectively. In these three figures, transcription factors (TF) are represented by squares, while other genes are represented by circles, and the identity of each one of these genes is presented in Appendix A.

In Figure 2 we can see the 29 genes form four disconnected sub-graphs. This indicates that there are four contrasting time expression profiles in the 29 genes annotated in *celcy*; the first groups four genes in the upper left hand side corner of the figure and follows that corner in clock-wise direction, a second group includes two genes, a third one includes 10 genes, and a fourth one includes 13 genes in the down left hand side corner.

In Figure 3, we see that the 10 genes of the *rep* GFN are organized into two disconnected sub-graphs. The largest, at the lower left hand side corner, contains eight interconnected genes linked by 28 connections, and the second, at the upper right side corner, is formed by only two genes with one connection.

In Figure 4 we see that the four genes of the *vir* GFN are organized into one fully connected graph, with all the six possible connections between those four genes.

#### 3.1.2. A Meta-Network

Having estimated GFN for three processes, *celcy*, *rep*, and *vir*, we can now note that those three networks share some common genes, and thus they can be linked to form a meta-network (MN). Figure 5 shows a schematic representation of the MN, where each GFN is represented by a single circle, and the number of genes shared by pairs of GFN is given at the links between GFN.

In Figure 5, we can appreciate the differences in size of the three GFN that form the MN, which include 29, 10, and four genes for the *celcy* (**C**), *rep* (**R**), and *vir* (**V**) GFN, respectively. We can also see that the three GFNs are inter-connected by sharing some genes between the GFN; six between *celcy* and *rep*, two between *celcy* and *vir*, and one between *rep* and *vir*. This last fact indicates that cell cycle (*celcy*), reproduction (*rep*), and response to virus (*vir*) in chili pepper are inter-dependent biological processes (BP).

Figure 2, Figure 3 and Figure 4 presented the graphs of the three GFNs, coloring genes in orange for the *celcy*, in golden yellow for the *rep*, and in blue for the *vir* GFN. Figure 6 presents the MN formed by the three GFNs, following that color scheme, but changing to white the color of genes shared by two or more of the GFN.

Before, in Figure 2 and Figure 3, we have seen that the GFN for *celcy* and *rep* included disconnected sub-networks; four for *celcy* and two for *rep*, while the GFN for *vir* presented a single fully connected graph (Figure 4). Disconnected sub-graph within a single GFN indicates that there are heterogeneous gene expression profiles, each one of them including a subset of the genes in the corresponding GFN.

In Figure 6, we can appreciate that the MN is formed by four disconnected sub-graphs, annotated with labels [P = 1] to [P = 4]. These four disconnected sub-graphs include highly correlated genes, which present characteristic gene expression profiles. Figure 7 presents the plots of the average standardized expression profiles (SEPs) on time for gene patterns [P = 1] to [P = 4], in panels (a) to (d), respectively. In these four panels, grey lines represent the average in the 12 accessions—which include two $F_{1}$ crosses between a D and a W genotypes [19], while red and blue lines represent the average in the six domesticated (D) and four wild (W) accessions, respectively. Thin vertical lines at each time point give the 95% confidence interval (CI) for the mean of standardized expression at each time and for each one of the groups.

Figure 7a,b show that there are differences between SEP from the domesticated (D, in red) and wild (W, in blue) accessions—the 95% CI are not overlapped in some fruit developing times, while Figure 7c,d do not show such differences between the D and W groups.

In summary, we found that the three GFN are inter-connected forming a MN (Figure 6), which consist of four disconnected sub-graphs, each one with a typical time expression profile (Figure 7). This implies that neither gene annotation per BP, nor expression profile in isolation, can fully explain the complexity of the phenomenon observed; for a deep understanding, we need to consider, in concurrence, both the BP annotation, as well as the particularities of typical time expression profiles. Appendix A presents the interpretation and biological implications for the development of chili pepper fruit of the four disconnected sub-networks that form the MN.

### 3.2. Finding Transcription Factors for Hub Genes

Transcription Factors (TF), in particular pioneer TF, are key regulators that initiate network changes [35]. In our GFN algorithm, only TF that are already among the genes initially input could be selected to form part of the output network. Nevertheless, using a similar approach to the one employed by the GFN algorithm, it is possible to obtain strong TF candidates to be regulating one or more of the genes that are already in a GFN. This is specially important for highly interconnected genes, i.e., “*hub genes*”, given that such TF could be playing a central role in the GFN regulation.

The algorithm “*Gene2TF*” to perform the selection of TF candidates is an adaptation of the *Gene2Gene* function, which calculates pairwise correlations between a target gene and all TFs in all genotypes available, using thresholds as the ones previously described to assure both, a significant co-expression, as well as robustness. Robustness is guaranteed by asking that the co-expression must be present in different genotypes. Details of the *Gene2TF* algorithm are presented and exemplified in Appendix A.

In the example presented with our *Capsicum* data, the only sub-network within the MN that does not include TF is the one with expression pattern [P = 2] within the *celcy* GFN (pattern [P = 2] in Figure 6). That sub-network includes 10 genes, which have a maximum expression at 10 DAA (Figure 7b); however, none of those genes is a TF. To find TF candidates to be controlling this sub-network, we ran the *Gene2TF* for every gene with expression pattern [P = 2], with the same thresholds than the ones used to obtain the GFN, and then we asked that the TF candidates must be present in the 12 genotypes available. This procedure selected 10 TF candidates for the genes with pattern [P = 2]. Furthermore, a new run of the GFN algorithm, including both the 10 genes with [P = 2], as well as the 10 TF candidates found, produced a highly connected and robust network, which is presented in Appendix A. However, experimental evidence will be needed to corroborate or discard the control role of those TF candidates on the genes with pattern [P = 2] within the *celcy* GFN; see, for example, [36,37,38].

### 3.3. Sensitivity Analyses

To evaluate the acuity of the *Gene2Gene* algorithm to input parameters, we designed a fully balanced grid on the combinations of the algorithm’s input thresholds, excepting the number of genotypes where the relations must be found, which was always set to the maximum level of stringency: 12 genotypes.

We ran the algorithm with the set of the *celcy* genes using the balanced grid of parameters, and we measured as output variables the number of genes and relations reported in the output. We found that the number of genes and relations reported by the algorithm have a strong linear correlation ($\hat{r}\approx 0.99;\;p<2\times 10^{-16}$), and the more important parameter to determine the number of genes and relations reported was the FDR threshold, followed by the minimum ${\hat{r}}^{2}$. The less important parameter that determines the numbers of genes and relations reported by the algorithm was the threshold to avoid regression outliers.

In summary, we conclude that the *Gene2Gene* algorithm allows the recovery of GFN, with consistency and flexibility for an ample set of reasonable combinations of stringency thresholds defined by the user. Details of the results of the sensitivity analyses are presented in Appendix A.

## 4. Discussion

Even when GCNs are powerful tools to disentangle the high complexity of the transcriptome, genome wide networks always result in graphs that are too large and complex for direct interpretation, as also happens with genome-scale metabolic networks [39]. In such large networks, it is possible to zoom in particular regions of interest, perhaps centered in a given gene, but, in that case, much of the information obtained will be wasted or not adequately noticed.

In contrast with the whole genome GCN, we advocate here for a bottom-up constructive approach, which begins with a small set of genes of interest and produces small and easy to interpret GFN. Furthermore, given that we ask that the relations must be repeatedly found in many genotypes, which ideally represent the diversity of the specie, the resulting GFN will have an evaluable level of robustness. We also argue that highly robust GFN are likely to present causal relations between genes because the majority of the not-causal relations, which could be either random or particular to a given genotype, are filtered out by the algorithm. Even when there are other statistical means to approach causality of GFN [40], we reason that repeatedly finding the same gene relation on independent experiments and genotypes gives the strongest rationale for causality. This common sense argument is analogous to the decision that could be taken about the fairness of a coin that repeatedly shows the same result when flipped; even when the coin could be fair, the likelihood of fairness compared with the likelihood of un-fairness rapidly decreases when we observe only one of the two possible results in many assays. In Appendix A, we further discuss the implication of our rule to infer causality for a gene relation.

To the best of our knowledge, our approach is the only one that explicitly takes into account repeatability in different genotypes to assure robustness of the inferred networks. For example, computational tools, such as WGCNA [41], DICER [42], CoXpress [43], or DiffCoEx [44], estimate networks, but without asking for repeatability in different genotypes so as to consider as valid each one of the relations found.

In the examples presented, we have seen that a single GFN could contain disconnected sub-graphs. The presence of such independent modules directly indicates that there are two or more dissimilar time expression profiles that share the same annotation, and that fact improves the understanding of the different processes within a single GFN.

On the other hand, we have also seen in the examples that two or more GFNs could be joined when they share genes, forming what, here, we call a “*Meta-Network*" (MN). Such joining process could proceed with as many GFNs as the researcher considers reasonable for a hierarchical ordering of BP. In the examples, we showed how the GFN obtained from the cell cycle (*celcy*), reproduction (*rep*), and response to virus (*vir*) BP could be joined in a single MN, whose biological relevance is discussed in Appendix A.

Furthermore, the *Gene2Gene* algorithm could be applied not only to a specific BP to obtain a GFN, but to any set of genes of particular interest of the researcher, for example, to genes annotated in an specific metabolic pathway, or to a set of lncRNA as done in [45], etc.

Besides, the *Gene2TF* algorithm allows the selection of TF candidates to be regulating any gene of interest (see Appendix A). Even when the correlation in time of expression of a target gene with a TF candidate in various genotypes could suggest the possibility of co-regulation [46,47], other bioinformatic or experimental means are needed to corroborate such a role [36,37,38].

One point worth discussing is the information given by genes that are excluded from a GFN when asking congruent correlation in a set of genotypes. For example, the first row in Table 1 shows that only approximately 8% (29/352) of the genes annotated in the *celcy* (cell cycle) BP are finally included in the GFN. This implies the existence of heterogeneity in the time expression patterns of the 352 − 29 = 323 excluded genes, which even when functionally annotated as related to cell cycle, present a diverse time expression profile among genotypes. Mining such specific time expression profiles could lead to a better understanding of the differences that exist among the set of tested genotypes—in this case with reference to cell cycle, but in general to any estimated GFN.

While in three out of four of the estimated sub-networks there is at least one TF that could be regulating expression (patterns [P = 1], [P = 3] and [P = 4] in Figure 6), in the fourth one (pattern [P = 2] in Figure 6), there is none. The inclusion of regulatory elements (TF) is of paramount importance to understand gene network changes [35], and thus all GFNs must include such elements—even when their functional role could be only putative. In this regard, the *Gene2TF* algorithm showed high efficiency by providing a set of 10 TF candidates that form a highly connected network with the genes previously found in the cell reproduction (*celcy*) GFN (Appendix A). These results give a solid working hypothesis to investigate the regulation of cell reproduction, a biological process of foremost practical, as well as theoretical importance, given its role in fruit size [48]—one of the key traits in fruit production.

As the final result of our method—applying the *Gene2Gene* and *Gene2TF* algorithms to the *Capsicum* data—we obtained a robust gene functional meta network (MN), linking three important biological processes and including 45 genes (Appendix A), of which 29 (64%) are TFs. This MN is easier to interpret than genome-wide networks because the researcher can focus on relations between specific processes of interest, with the additional advantage that sub-networks show particular time expression profiles (Figure 7). This approach can be directly applied using the *Salsa* software to other sets of *Capsicum* genes during fruit development, grouped by different criteria, or can also be implemented for other organisms or development processes by following the guidelines given in [20].

About the design of future experiments using our methodology, the *Gene2Gene* and *Gene2TF* algorithms are only applicable to development processes for which time expression profiles exist in a set of different genotypes. The first question is how many time points need to be fixed to estimate time expression profiles—which can always be transformed to SEPs [20]. In this regard, any case with less than five time points will be too flimsy to obtain useful correlation coefficients, and of course, more time points will give a more precise discrete description of the continuos changes of gene expression through time. It is important to take into account that each time point within each genotype will demand the construction and sequencing of a minimum of two RNA-Seq libraries.

The second issue in this regard concerns the nature and number of the genotypes to be used. In any real case, it will be impossible to include all the relevant genetic diversity present in a crop or in a model organism. However, usefulness of the method will be maximized by selecting the largest number of almost independent and contrasting genotypes. In our case, we selected six D chili pepper accessions with contrasting fruit characteristics, four W accessions presenting also diversity in fruit as well as in geographic origin, and two crosses between a D and a W accessions to include $F_{1}$ hybrids.

The function “pred.error.g2g()” in the *Salsa* package calculates the predicted or realized error in finding a correlation between two genes significant at a fixed false discovery rate (FDR). By exploring a grid of values for the total number of accessions available, the number of genotypes where the relation must be found to be considered in the GFN and the maximum FDR that could be tolerated, the researcher can reach a reasonable decision on the number of genotypes to be employed (see Appendix A).

## 5. Conclusions

The *Gene2Gene* and *Gene2TF* algorithms, now implemented and demonstrated in the R package *Salsa* (version 1.0), provide powerful tools to study gene co-expression of gene sets in a hierarchical way. When there are gene expression profiles from a representative set of genotypes, these tools allow the estimation of robust and easy to interpret GFNs (see also Appendix A).

We expect that the research community interested in fruit development (or other development processes) will adopt our methodology. We also anticipate that the results and functions already present in the *Salsa* R package can lead to further biologically relevant findings. 

## Figures and Tables

**Figure 1 plants-12-01148-f001:**
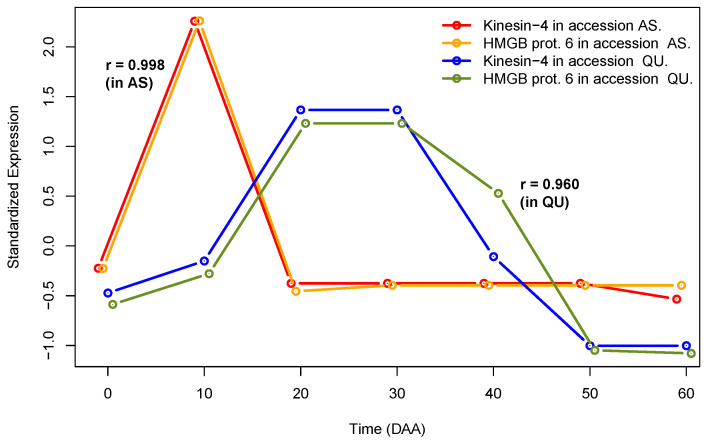
Example of SEPs for two highly correlated genes in two independent genotypes.

**Figure 2 plants-12-01148-f002:**
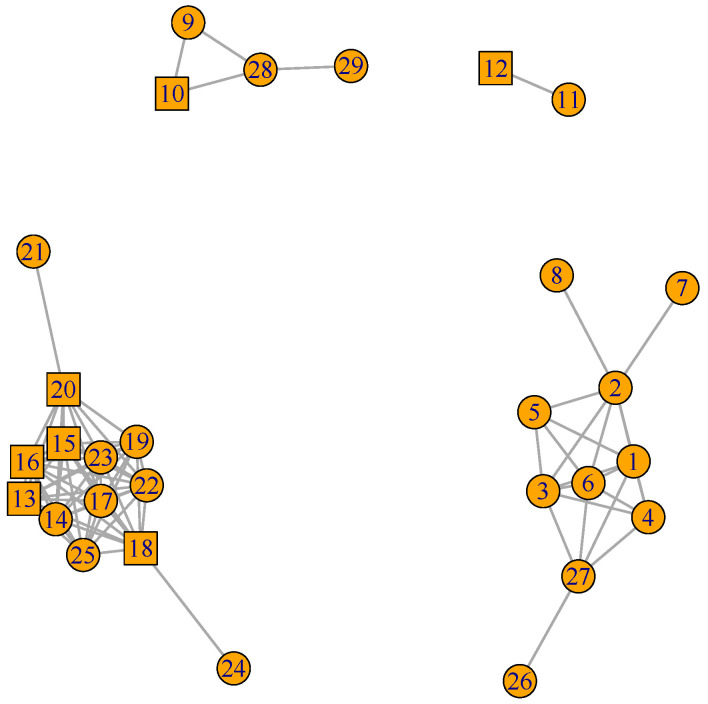
Cell Cycle (*celcy*) GFN. 29 genes and 81 pair connections in four disconnected sub-graphs. TF represented by squares, while other genes are represented by circles (Appendix A for gene descriptions).

**Figure 3 plants-12-01148-f003:**
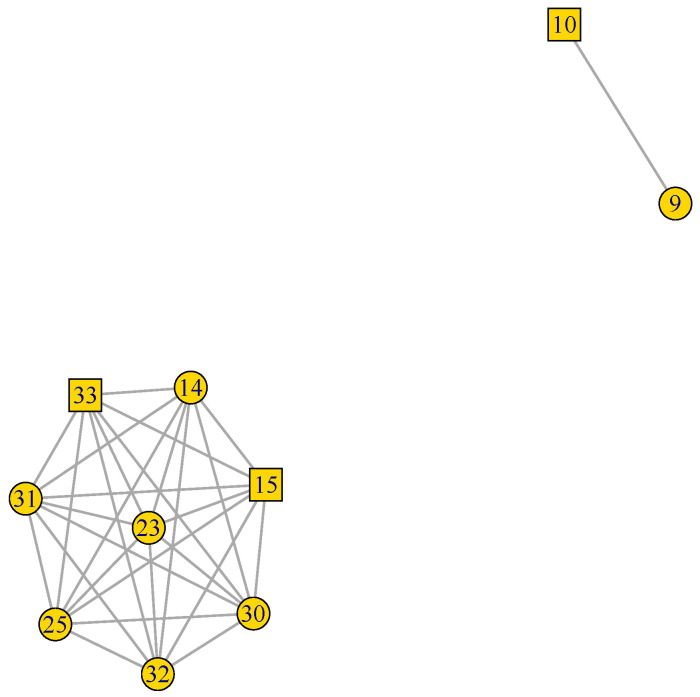
Reproduction (*rep*) GFN. 10 genes and 29 pair connections in two disconnected sub-graphs. TF represented by squares, while other genes are represented by circles (Appendix A for gene descriptions).

**Figure 4 plants-12-01148-f004:**
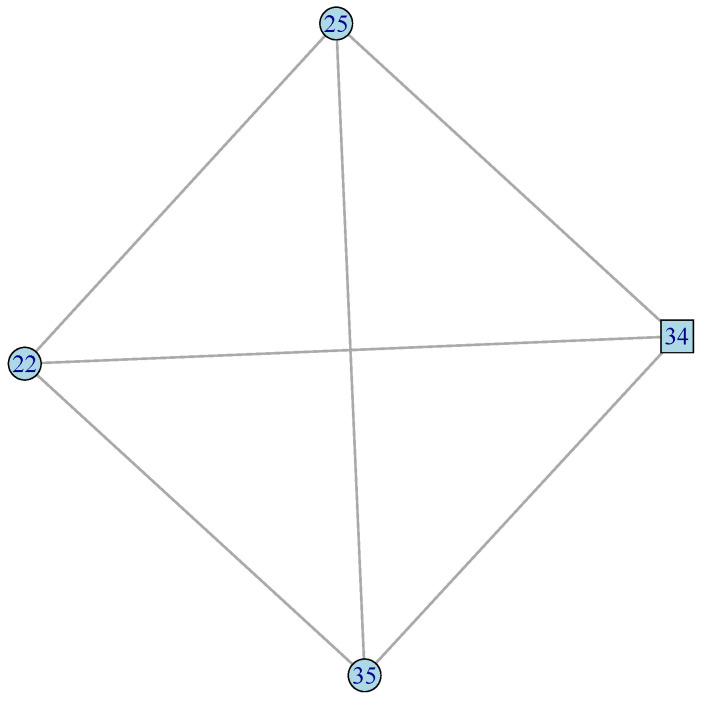
Response to virus (*vir*) GFN. Four genes and six connections in a fully connected graph. TF is represented by squares, while other genes are represented by circles (Appendix A for gene descriptions).

**Figure 5 plants-12-01148-f005:**
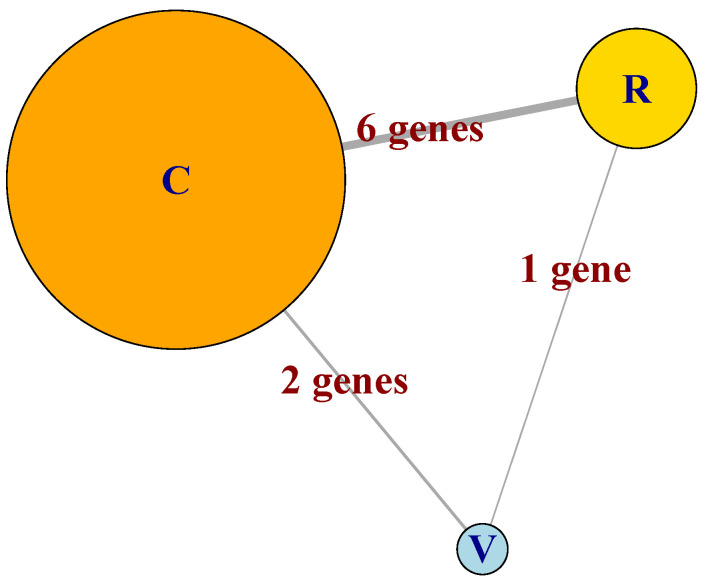
Schematic representation of the MN formed by the GFN *celcy*, *rep* and *vir*, represented as the circles labeled as **C**, **R** and **V**, respectively. Sizes of circles are proportional to the number of genes at each GFN and links are annotated with the number of genes shared between pairs of GFN.

**Figure 6 plants-12-01148-f006:**
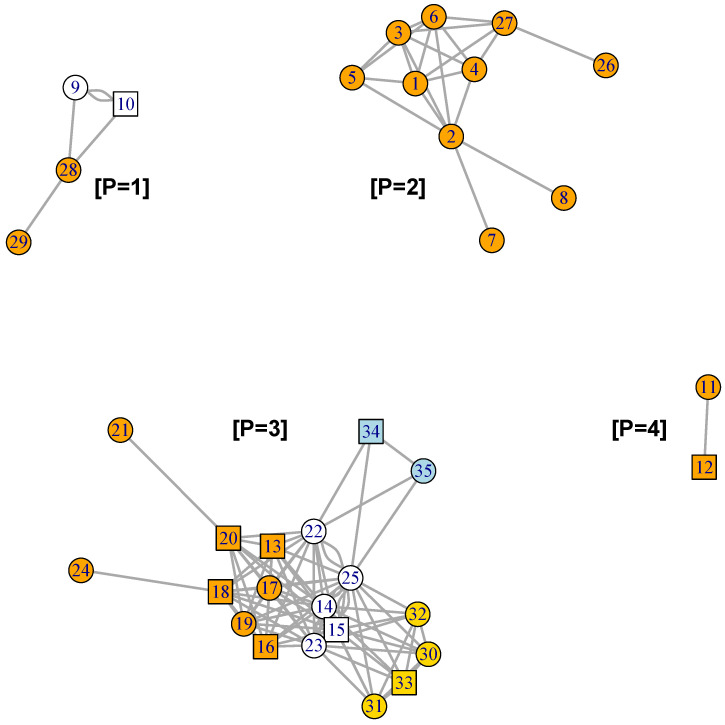
Graph of the MN formed by the 3 GFN. Genes found exclusively in BP *celcy* are colored in orange, the ones exclusively in *rep* are colored in golden yellow, and the ones exclusively in *vir* are colored in blue, while genes shared by two or more BP are in white. Disconnected sub-networks are annotated with labels **[P = 1]** to **[P = 4]** (Appendix A for gene descriptions).

**Figure 7 plants-12-01148-f007:**
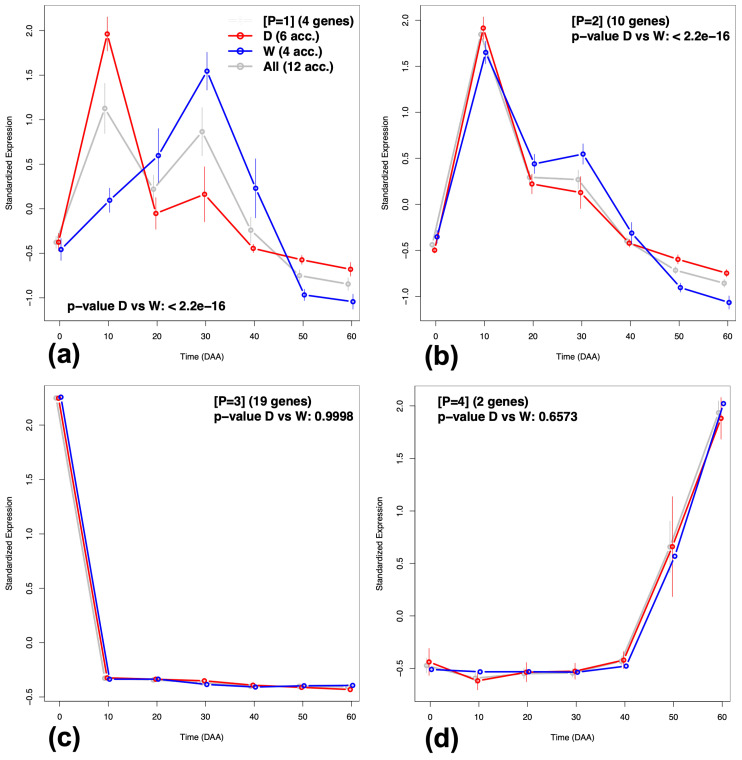
Average Standardized Expression Profiles (SEPs) patterns for genes included in the Meta-Network in Figure 6. (**a**) Pattern **[P = 1]**. (**b**) Pattern **[P = 2]**. (**c**) Pattern **[P = 3]**. (**d**) Pattern **[P = 4]**.

**Table 1 plants-12-01148-t001:** Summary of results for GFN estimation.

BP	In	Out	% Out/In	Con.	$\bar{r}$	$\bar{q}$
*celcy*	352	29	8	81	0.9859	0.00729
*rep*	228	10	4	29	0.9934	0.00257
*vir*	33	4	12	6	0.9936	0.00126

BP—Abbreviation for the Biological Process, In—Number of genes in the input, Out—Number of genes in the output, Con.—Number of connections in the GFN, $\bar{r}$—Average value of $\hat{r}$ in all connections. $\bar{q}$—Average value of $\hat{q}$ in all connections.

## Data Availability

Original RNA-Seq data have been deposited into NCBI’s Gene Expression Omnibus and are accessible through the GEO Series accession number GSE165448: https://www.ncbi.nlm.nih.gov/geo/query/acc.cgi?acc=GSE165448 (accessed on 27 February 2023). All curated data and functions for the analyses and algorithms are publicly available in the R package “*Salsa*” (version 1.0): https://zenodo.org/record/4767445#.YmAHvZPMLaY (accessed on 27 February 2023).

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
