# Peer review of "Gene Functional Networks from Time Expression Profiles: A Constructive Approach Demonstrated in Chili Pepper (Capsicum annuum L.)"

_plants, 2023, doi:10.3390/plants12051148_

Round 1

Reviewer 1 Report

The manuscript “Gene Functional Networks from Time Expression Profiles: A constructive approach demonstrated in chili pepper” by Flores-Díaz et al sent for publication to Plants deals with important topic such as construction of a gene co-expression networks based on transcriptomic data. The present work will be of high interest to the scientific community working in that area. The manuscript is well written and I don’t have any additional comments. Overall, the manuscript deserved to be published in the present form.

Author Response

Dear Reviewer number 1,

Thank you for your evaluation of our manuscript. We acknowledge your assessment and attach here the file "mainGFNchanged.pdf", which includes -underlined in orange, the minor changes of the new version of the manuscript, which are the result of the recommendations of other two Reviewers. Given that the changes are only minor, we expect that you will agree with the new version.

With kind regards,

Octavio Martínez.
Corresponding author.

Reviewer 2 Report

The authors have applied the gene co-expression networks to understand functional interactions among genes and to further provide novel insights. In particular, the authors have developed an algorithm to construct Gene Functional Networks for genes annotated in a given biological process of interest based on the assumption that there are genome-wide time expression profiles for a set of representative genotypes of the species under investigation. Specifically, the method is based on the correlation of time expression profiles. The novelty of this method included that a gene expression relation must be repeatedly found in a given set of independent genotypes to be considered valid, automatically discarding the relations particular to specific genotypes to assure a network robustness which can be set a priori. This method was tested extensively based on transcription factor candidates regulating hub genes within a network and the gene expression during the development of the fruit in a diverse set of chili pepper genotypes.

I believe that the authors have provided sufficient background information, explained well the methodologies and algorithms, presented the data with appropriate tables and figures, and concluded appropriately based on data available. I have no major technical concerns but some minor suggestions on the presentation of the texts, as I have listed below. I believe these changes could significantly improve the overall presentation of this manuscript, if a revision is requested by the editor.

Lines 45-48: these three properties should be described with explanations in the text but not listed as such.

Lines 79 and after: the authors should indicate explicitly the goals of their study.

Line 386: what is the point to establish this section 4.1. of the entire Discussion? As a matter of fact, it would be ideal to establish a few subsections in Discussion to focus on a few topics with in-dep discussion of the large amount of data presented in Results.

Author Response

Dear Reviewer number 2,

Thank you for your evaluation of our manuscript. In what follows we answer one by one your kind suggestions, and attach here the file "mainGFNchanged.pdf", which includes -underlined in orange, the lines changed in the new version.

1) Lines 45-48: these three properties should be described with explanations in the text but not listed as such.
Answer: We modified those lines. Lines 43 and 44 in the new version present the characteristics that we consider for a GFN. As in the old version, the following paragraphs explain in detail the meaning of such characteristics.

2) Lines 79 and after: the authors should indicate explicitly the goals of their study.
Answer: Lines 76 to 77 in the new version explicitly indicate the main goal of our work. Additionally, lines 87 to 91 in the new version give the secondary goal of the work.

3) Line 386: what is the point to establish this section 4.1. of the entire Discussion? As a matter of fact, it would be ideal to establish a few subsections in Discussion to focus on a few topics with in-depth discussion of the large amount of data presented in Results.

Answer: We agree in that to establish section "4.1 Designing future experiments" in the discussion was unnecessary. Instead, we added two new paragraphs (without sub-heading) giving a brief but in-depth discussion of our main results. Such new paragraphs are now in lines 390 to 400 and 401 to 409, respectively. The paragraphs in the old "4.1" section are now on lines 410 to 431, but without sub-heading.

We hope that this new version of the manuscript, which also includes minor changes suggested by another reviewer, will be acceptable for publication.  

With kind regards,

Octavio Martínez.
Corresponding author.

Reviewer 3 Report

Review of plants-2222623

Gene Functional Networks from Time Expression Profiles: A constructive approach demonstrated in chili pepper

Alan Flores-Díaz, Christian Escoto-Sandoval, Felipe Cervantes-Hernández, José J. Ordaz-Ortiz, Corina Hayano-Kanashiro, Humberto Reyes-Valdés, Ana Garcés-Claver , Neftalí Ochoa-Alejo and Octavio Martínez

The authors developed an algorithm for constructing gene functional networks based on comparing the expression profiles of genes annotated as belonging to the same biological process over the same time course conducted under the same environmental conditions in multiple accessions of the same species. The rationale is that if the same pattern is observed in all accessions over the same time course then it must be significant.  They also developed an algorithm for identifying candidate transcription factors that might be regulating these gene functional networks. They provide examples of how their algorithms were used to identify gene functional networks and transcription factors that might regulate these networks using RNAseq data from fruit development in 12 accessions of chili pepper.

The algorithms provide an approach to identifying gene functional networks that is more user-friendly than the current whole genome approaches.  They are therefore worth sharing with the plant community.

Just a few minor quibbles.

Please include the latin binomial for chili pepper in the title.

Lines 90-92: Please rewrite for clarity

Lines 307-315: Please rewrite for clarity

Author Response

Dear Reviewer number 3,

Thank you for your evaluation of our manuscript. In what follows we answer one by one your kind suggestions, and attach here the file "mainGFNchanged.pdf", which includes -underlined in orange, the lines changed in the new version.

1) Please include the latin binomial for chili pepper in the title.

Answer: The title of the manuscript was modified.

2) Lines 90-92: Please rewrite for clarity.

Answer: We rewrote lines 90-92. The corresponding lines in the new version are now lines 87-91 (we also followed the suggestion of another reviewer in the same sense). We hope that the new lines give a clearer explanation.

3) Lines 307-315: Please rewrite for clarity 

Answer: We modified for clarity lines 307-315. The paragraph in the new version is in lines 309-319.

We hope that this new version of the manuscript, which also includes minor changes suggested by another reviewer, will be acceptable for publication.  

With kind regards,

Octavio Martínez.
Corresponding author.
